# Antioxidant Molecules from Marine Fungi: Methodologies and Perspectives

**DOI:** 10.3390/antiox9121183

**Published:** 2020-11-26

**Authors:** Giovanni Andrea Vitale, Daniela Coppola, Fortunato Palma Esposito, Carmine Buonocore, Janardhan Ausuri, Emiliana Tortorella, Donatella de Pascale

**Affiliations:** 1Institute of Biochemistry and Cell Biology (IBBC), National Research Council, Via Pietro Castellino 111, 80131 Naples, Italy; giovanniandrea.vitale@ibbc.cnr.it (G.A.V.); carmine.buonocore@ibbc.cnr.it (C.B.); janardhan.ausuri@ibbc.cnr.it (J.A.); emiliana.tortorella@ibbc.cnr.it (E.T.); 2Department of Marine Biotechnology, Stazione Zoologica Anton Dohrn, Villa Comunale, 80121 Napoli, Italy; daniela.coppola@szn.it (D.C.); fortunato.palmaesposito@szn.it (F.P.E.); 3Institute of Biosciences and BioResources (IBBR), National Research Council, Via Pietro Castellino 111, 80131 Naples, Italy

**Keywords:** antioxidants, fungi, marine natural products, extraction protocols, applications

## Abstract

The marine environment represents a prosperous existing resource for bioprospecting, covering 70% of the planet earth, and hosting a huge biodiversity. Advances in the research are progressively uncovering the presence of unknown microorganisms, which have evolved unique metabolic and genetic pathways for the production of uncommon secondary metabolites. Fungi have a leading role in marine bioprospecting since they represent a prolific source of structurally diverse bioactive metabolites. Several bioactive compounds from marine fungi have already been characterized including antibiotics, anticancer, antioxidants and antivirals. Nowadays, the search for natural antioxidant molecules capable of replacing those synthetic currently used, is an aspect that is receiving significant attention. Antioxidants can inactivate reactive oxygen and nitrogen species, preventing the insurgence of several degenerative diseases including cancer, autoimmune disorders, cardiovascular and neurodegenerative diseases. Moreover, they also find applications in different fields, including food preservation, healthcare and cosmetics. This review focuses on the production of antioxidants from marine fungi. We begin by proposing a survey of the available tools suitable for the evaluation of antioxidants, followed by the description of various classes of marine fungi antioxidants together with their extraction strategies. In addition, a view of the future perspectives and trends of these natural products within the “blue economy” is also presented.

## 1. Introduction

Nowadays, the research has focused the attention on microorganisms for the development of novel natural products. The use of microorganisms represents an excellent opportunity of obtaining a continuous source of bioactive compounds, so as to meet the demands of industries, that require large quantities of biomass for clinical testing and/or industrial production [1].

The redundancy in the discovery of known compounds, generally produced by already known species, is increasing the need for the discovery of new microorganisms producing alternative drugs; in this contest, marine microorganisms are becoming a top spot for natural products discovery, as they are a largely untapped source. 

The marine environment covers more than 70% of Earth’s surface and represents the largest ecosystem on Earth, characterized by extremely variable and hostile physico-chemical parameters (low temperature, limited light access, high salinity, and high pressure). The world’s oceans and seas account about 90% of the living biomass of our planet, dominated by unicellular microorganisms [2]. In order to face the wide range of ecological niches that characterize the oceans, marine microorganisms have evolved across the evolutionary time scale, unique metabolic and genetic adaptation that led to an extraordinary secondary metabolites diversity, with unmatched structures [3,4,5]. Recently, it was even understood that many bioactive molecules previously isolated from marine macroorganisms, are actually produced by associated microorganisms [6,7]. Moreover, the fraction of Food and Drug Administration (FDA) approved marine natural products has also become enriched with microbial compounds, which represent a significant portion of approved antibiotics [8]. 

Although microorganisms have an essential role in the ecosystem functioning, and global biogeochemical cycles of the major elements [9], we still know little about marine microbial diversity, the ecological role in the ecosystem, and their metabolic capabilities. The development of genomic approaches made possible to identify marine microorganisms, allowing scientists to understand the full extent of microbial biodiversity and functions in the world’s oceans [10].

Marine microbial eukaryotes, among which fungi, are abundant and ecologically important members of marine microbiota, and many studies, based on molecular and metagenomics approaches, have documented their once-unimaginable diversity [11,12], even in deep-sea hydrothermal ecosystems [13]. In fact, it is noted that marine fungi are widely distributed in marine environments, from shallow water to deep sea, even down to the polar ice covers. They are found in sediments and in all kinds of living and dead organic matter. Cultivation-dependent studies demonstrated the huge abundance of fungi in marine macroorganisms, like sponges and algae [14,15,16,17,18,19]. It has been observed that most marine sponges harbor often fungi with representatives of *Acremonium*, *Aspergillus*, *Fusarium*, *Penicillium*, *Phoma*, and *Trichoderma* [18,20]. Moreover, thanks to the accumulation of fungi in sponges, these macroorganisms have been also used to find fungi belonging to the less common taxa, such as *Beauveria*, *Botryosphaeria*, *Epicoccum*, *Tritirachium*, *Paraphaeosphaeria*, *Bartalinia* and *Volutella* [16,21].

The marine fungi represent an important opportunity for bioprospecting, as the exploration of their potential in terms of the exploitation of molecules is still at an early stage. In order to evaluate the metabolic potential of marine fungi, several genomic sequences of fungi are currently under way. The already completed sequences showed a huge biosynthetic capacity of fungi, with generally approximatively 30 to 40 biosynthetic gene clusters coding for secondary metabolites in every genome [22,23,24]. Therefore, marine fungal biotechnology represents an exciting and promising area of investigation [25], that is increasing. 

Recently, the number of new marine natural products is definitely high, e.g., in 2016 alone, 1277 new compounds with promising biomedical applications were identified, and 36% of these molecules are produced by marine fungi [3]. Only 15 molecules were identified from fungi by 1992 [26] and around 270 until 2002 [14]. Approximately 100 new marine fungal compounds were described from 2000 to 2005 [27] and 690 metabolites were listed by 2010 [15]. This number is still increasing, and in 2016 are reported 328 new compounds from marine fungi not associated with mangroves, and 142 from mangrove-associated fungi (Blunt et al., 2018).

Marine fungi produce a plethora of well-known bioactive compounds, among which are anticancer, antibiotic, antiangiogenesis, and antiviral compounds, and molecules having antiproliferative activity [3]. Interestingly, a large number of antioxidants molecules are reported which, due to their unique properties, can be used in a variety of fields, including food, cosmeceuticals, and pharmaceuticals. 

Free radicals and oxidants have a dual role in the body, as they can be both toxic and beneficial compounds. They are produced during the normal cell metabolisms, but also induced by external sources, e.g., pollution, cigarette smoke, radiation, medication. When these molecules are gradually accumulated, due to the imbalance between reactive oxygen species (ROS) generation and antioxidant defence systems, they generate oxidative stress. The ROS family includes numerous types of molecules that cause harmful effect towards cellular constituents. They react with most biological macromolecules (lipids, proteins, and nucleic acids) causing their degradation and destruction, with different huge consequences on cell biology. Several evidences in experimental and human trials indicate that oxidative stress plays an important part in the pathogenesis of various chronic and degenerative processes, such as cancer, autoimmune disorders, aging, cataract, rheumatoid arthritis, inflammation, cardiovascular and neurodegenerative diseases [28,29,30]. 

Moreover, lipid oxidation continues to represent a major problem in food industry. It may occur in foods during harvesting, processing, and storage, (i) destroying organoleptic properties (with the development of repugnant flavours and odours), or valuable nutrients (e.g., essential fatty acids, lipid-soluble vitamins, and proteins), and (ii) producing toxic compounds, thus making the lipid or lipid-containing foods unsuitable for consumption [31]. In particular, lipid oxidation is also responsible for quality deterioration of muscle foods and dairy products, as well as fruits and vegetable crops [32,33].

Study on molecules with antioxidant effects is a promising strategy for prevention and treatment of these processes. For this reason, nowadays, there is increasing interest to study the properties of agents currently present, and to investigate the possibility of using novel antioxidant compounds.

This review is an overview on known antioxidants produced by marine fungi so far. We first deal with antioxidants deactivation mechanisms of reactive species and with the available assays to detect them, then we discuss the molecules with antioxidant activity, and the extraction methodologies, evaluating their potential applications in different application fields.

## 2. Radical Interactions and Antioxidant Properties Assessment

### 2.1. Radicals Formation and Antioxidant Action 

Most of the potentially harmful effects of oxygen are due to the formation and activity of a number of chemical compounds, known as ROS, which are mostly free radicals, including superoxide (O_2_•−), hydroxyl (•OH), peroxyl (RO_2_•), and alkoxyl (RO•) that have the tendency to extract an electron from other molecules, and non-radicals that are either oxidizing agents or molecules easily converted into radicals, such as hypochlorous acid (HOCl), ozone (O_3_), singlet oxygen (^1^O_2_), and hydrogen peroxide (H_2_O_2_). 

A free radical can be defined as any molecular species capable of independent existence that contains an unpaired electron in an atomic orbital and this condition make many radicals unstable and highly reactive. They can either donate an electron to or accept an electron from other molecules, therefore behaving as oxidants or reductants. 

Their production seems to be part of a complex signalling web where they play a regulatory role, as mediators in several physiological processes. In particular, ROS are formed as a natural by-product of the primary metabolism and have important roles for the cell to balance redox homeostasis and modulate the cellular activities such as cell survival, stressor responses, and inflammation [34,35].

Every radical reaction presents (i) an initiation phase in which the radical is generally formed through an homolytic bond cleavage, (ii) a propagation phase in which radicals steal an electron from a surrounding molecule producing another radical, and (iii) a termination step which occurs when two radicals quench together deactivating each other. Unfortunately, radical terminations are not frequent events due to thermodynamic and kinetic factors, while their uncontrolled and massive propagation in the cell can trigger radical cascades making the phenomenon uncontrollable. 

There are numerous endogenous (mitochondria, peroxisomes, endoplasmic reticulum, phagocytic cells, etc.) and exogenous (pollution, alcohol, tobacco smoke, heavy metals, transition metals, industrial solvents, pesticides, certain drugs) factors which contribute to the generation of free radicals [36].

Antioxidant molecules represent the main defence system against ROS. They can act at different defence levels for example by preventing free radical formation (first line of defence), by scavenging them (second line of defence), by repairing the damages(third line of defence), by generating cell adaptation to the oxidative signals (fourth line of defence) [37].

There are several types of antioxidants which can be found in humans, animal, plants and microorganisms which can be used as defence system towards oxidative stress. 

A difference occurs between enzymatic and nonenzymatic antioxidants. In enzymatic antioxidants, a detoxification pathway is constituted by multiple enzymes which first intervene to convert superoxide into hydrogen peroxide that than is further reduced to give water. Superoxide dismutase (SOD), Catalase (CAT), Glutathione (GSH) systems are all enzymes involved in the detoxification of ROS.

The non-enzymatic antioxidants take part in the first and in the second line of defence inhibiting the formation of new reactive species, by interacting with the transition metal ions or inactivating oxidants and radicals [38]. Among non-enzymatic antioxidants there are polyphenols, carotenoids, vitamins, glutathione, uric acid and other compounds.

Effective antioxidants are radical scavengers that break down radical chain reactions and, based on their chemical nature, they can display different mechanisms, including ^1^O_2_ quenching, radical scavenging single-electron transfers, hydrogen transfer, and metal chelation [39]. Several low-molecular weight antioxidants are known to exert their activity mainly by interacting with ROS neutralizing them or giving rise to more stable radicals, this feature is crucial for antioxidants [36,40,41]. For instance, it is well-known Vitamin E is a lipid-soluble antioxidant involved in the termination of lipid peroxidation, in fact it can protonate two peroxyl radicals with the formation of a stable adduct. Differently, ascorbic acid is a well-known hydrophilic antioxidant able to scavenge several radical including O_2_^•–^ and ONOO^-^, to synergistically work together with α-tocopherol terminating lipid peroxidation, and together with GSH to regenerate α-tocopherol [42,43,44]. 

### 2.2. Antioxidant Assays 

There are several in vitro and in vivo tests to evaluate the antioxidant activity of compounds of interest and can be divided into two general groups based on the reaction mechanisms involved in the reduction process. The in vitro method is off two groups, the first group of method is based on the single electron transfer (SET) and the second group is based on the hydrogen atom transfer (HAT) [45]. Both lead to the inactivation of free radicals but by different kinetics and secondary reactions. The HAT reaction is a concerted movement of a proton and an electron in a single kinetic step. In HAT mechanisms, the free radical removes one hydrogen atom from the antioxidant, and the antioxidant itself becomes a radical. The SET reaction is imitated by single-electron transfer from the nucleophile to the substrate, producing a radical intermediate, whose fate can then be involved in any number of events. In SET mechanisms, the antioxidant provides an electron to the free radical and itself then becomes a radical cation. It is very difficult to distinguish between HAT and SET reactions. In most situations, these two reactions take place simultaneously, and the mechanism of the reaction is determined by the antioxidant’s structure and solubility, the partition coefficient and solvent polarity. Examples of HAT-based assays include ABTS [2,2′-azinobis-(3-ethylbenzothiazoline-6-sulfonate)] and oxygen radical absorbance capacity (ORAC). Examples of SET-based assays include 2,2-diphenyl-1-Picrylhydrazyl (DPPH) and Ferric reducing antioxidant power (FRAP) [46]. 

The capacity of antioxidants in vivo against oxidative stress is determined by several factors. Above all, the bioavailability, that is, absorption, transportation, distribution, retainment, metabolism, and excretion of antioxidants, is one of the major factors in determining the capacity in vivo. The antioxidant activity in vivo may be assessed by the effects of antioxidants on the level of oxidative stress in animal models or in biological fluids, and tissues such as plasma, erythrocytes, urine, and cerebrospinal fluids from humans and experimental animals. Saliva and tear may also be used as non-invasive samples [47]. Among the in vivo tests there are SOD, CAT, Gamma Glutamyl Transferase (GGT), GSH assays.

It is almost impossible to compare antioxidant methods with one another [48], generally it is necessary to consider the source of ROS as well as the target substrate before setting up the best experimental test. In this review we will focus only on in vitro assays, as rapid, low-cost and reliable tests to detect the presence of antioxidant activity in marine fungal extracts, fractions and purified compounds.

#### In Vitro Antioxidant Assays

There are several in vitro methods for evaluating antioxidant activities (Table 1). Generally, in vitro antioxidant tests should be fast, cheap, and reliable in order to allow the selection of the most effective antioxidant molecule [49,50]. The choice of a select test depends by different factors, especially by the nature of the target antioxidant molecule. 

Among SET in vitro assays, the DPPH method is probably the most popular one to detect antioxidant activity. The molecule 1,1-diphenyl-2-picrylhydrazyl (α,α-diphenyl-βpicrylhydrazyl) consists of stable free radical resembling the structure of free radical [48]. The DPPH test is based on the ability of the stable 2,2-diphenyl-1-picrylhydrazyl free radical to react with hydrogen donors and change from purple to yellow. It has a maximum absorption at 519 nm in ethanol, and the change in optical density can be monitored by spectrophotometer. An advantage of the DPPH assay is that it is an easy, economic, and rapid method to evaluate the radical scavenging activity of non-enzymatic antioxidants. The biggest limitation of the DPPH assay is that it is not related to specific free radicals that have physiological relevance [51].

Another colorimetric assay is the ABTS assay. ABTS*+ [2,2-azino-bis (3-ethylbenzthiazoline-6-sulfonic acid)] is a stable radical with a blue-green chromophore showing maximum absorption spectra at 734 nm in water. When the antioxidant molecule is treated with this compound, it is decolorized indicating the antioxidant capability of the molecule of interest [48]. 

The results obtained by DPPH and ABTS assays are usually comparable but in certain cases one test can be preferred to the other. For example, the ABTS method can be used at different pH levels (unlike DPPH, which is sensitive to acidic pH) and thus is useful when studying the effect of pH on antioxidant activity of various compounds [46]. Additionally, ABTS is soluble in aqueous and organic solvents and is thus useful in assessing antioxidant activity of samples in different media. In general, highly pigmented and hydrophilic antioxidants are better assessed for antioxidant activity using the ABTS assay compared to the DPPH assay [52].

Another kind of colorimetric method is the FRAP assay (SET method), based on the ability of antioxidants to reduce ferric ions. FRAP assays are widely used to detect the total antioxidant potential by reduction of complexes 2,4,6-tripyridyl-s-triazine (TPTZ) with ferric chloride hexahydrate (FeCl_3_·6H_2_O). The solution will turn brownish forming blue ferrous complexes confirming the antioxidant potential of the extract. The reaction is monitored by measuring the change in absorption at 593 nm. FRAP method has its own limitations, particularly for measurements below non-physiological pH values 3.6. This method also cannot detect slow-reacting polyphenolic compounds and thiols [53]. Not all antioxidant molecules will reduce Fe^3+^ at the same pace and changes in absorbance are also observed even after 10 min (incubation time) [45]. The prolongation of incubation time could give reliable results. Moreover, FRAP cannot detect species that act by radical quenching (H transfer), particularly antioxidants containing a thiol group, such as glutathione and proteins.

CUPRAC (Cupric ion reducing antioxidant capacity) is based on the redox reaction between antioxidant and free radicals with the measurement of reduction of cupric ions to cuprous ions. This is a simple, reliable, and low-cost method which provides extensive information about the reducing ability of the antioxidant fungal extract. Neocuroine, a methylated phenanthroline derivative is used for this purpose. This chelates the metal copper and aids in formation of bis(neocuproine)copper (II) chloride. CUPRAC assay is advantageous over other assays as they yield stable products. Coloured reagents are light, pH, and humidity sensitive, whereas cupric reagent-coloured chelate of Cu(I)-Nc is found to be stable and insensitive to other such external factors [45].

Among HAT-based assays, ORAC assay is used to evaluate the capacity of antioxidant compounds that scavenge free radicals. It employs a fluorescent probe and a free radical generating compound. When antioxidant is combined with this mixture, the resulting loss of fluorescence correlates with the antioxidant capacity of the molecule [45]. β-phycoerythrin (β-PE) is generally employed as fluorescent probe that generates free radicals in reaction with 2,2′-azobis (2-amidino-propane) dihydrochloride (AAPH). An efficient antioxidant compound will inhibit the loss of fluorescence and it is monitored by microplate fluorescence reader. Some limitations of using β-phycoerythrin regard the problem of photo-bleach under plate-reader conditions and interactions with polyphenol compounds during the incubation. Naguib et al. suggested Fluorescein (3′,6′-dihydroxy-1-oxospiro(2-benzofuran-3,9′-xanthene)-5-carboxylic acid), a synthetic nonprotein probe which has been improvised in overcoming limitations of β-PE and product pattern is in line with hydrogen atom transfer mechanism [54]. 

In the Hydroxyl radical scavenging assay, the radical is generated when hydrogen peroxide reacts with Fe (II) (Fenton reaction). Hydroxyl radical is one of the common ROS that causes damage to cell by reacting with polyunsaturated fatty acid moieties of the cell membrane. Hydroxyl radical scavengers can protect deoxyribose, benzoate, or salicylate from being damaged by hydroxyl radicals and the reaction can be followed by spectrophotometer. However thus the mixture reaction needs iron, and as many antioxidants are also metal chelators, it is difficult to discriminate if the reaction is due to an antioxidant or to a metal chelation activity [55].

TBARS (Thiobarbituric acid reactive substance) assay measures the inhibition of production of TBARS from sodium benzoate under the influence of free oxygen radicals derived from Fenton’s reaction [48]. It is a well-known method for screening and monitoring lipid peroxidation. A standard solution of Fe-EDTA complex reacts with hydrogen peroxide by a Fenton reaction, leading to the generation of hydroxyl radicals. The ROS generated degrade benzoate and the antioxidant activity is demonstrated by red, fluorescent adduct which is measured spectrophotometrically at 532 nm [56]. 

Lipid peroxidation inhibition capacity (LPIC) assay measures the ability of both lipophilic and hydrophilic antioxidants to protect a lipophilic fluorescent probe. It is based on the reaction of a chromogenic reagent, N-methyl-2-phenylindole, with malondialdehyde (MDA) and HAE (4 -hydroxyalkenals) at 45 °C. The reaction leads to the formation of a stable chromophore (carbocyanine dye) with maximal absorbance at 586 nm [57].

Nitric oxide free radical scavenging assay. Nitric oxide radical (NO) is a highly RNS produced by oxidation of terminal guanido-nitrogen atoms of L-arginine in organism. Overproduction of such RNS leads to a state called nitrosative stress. Nitrosative stress can cause damage to membrane fatty acids and leads to cell oxidative damage. Nitroprusside is added in phosphate saline buffer with Griess reagent and the oxidant molecule/extract, and the reaction is followed spectrophotometrically evaluating the % of scavenging activity [45,58].

Metal chelating activity. Ferrozine is a highly ferrous-stabilizing ligand. Ferric ion in presence of Ferrozine forms a red colour Fe(II)-Fz complex which decreases its colour in presence of metal chelators [59]. In this method, the antioxidant activity can be followed by measuring the absorbance variation using spectrophotometer [60].

**Table 1 antioxidants-09-01183-t001:** List of commonly employed antioxidants invitro assays.

Antioxidant In Vitro Assay ^a,b^	Outcome of the Assay	Reference
DPPH assay	Production of reduced DPPH (Colorimetry)	[48]
2,2′-azino-bis (3-ethylbenzthiazoline-6-sulphonic acid) ABTS*+	Production of reduced ABTS	[61]
Ferric reducing antioxidant power (FRAP)	Colorimetric method. Production of reduced Fe^2+^ (TPTZ (2,4,6-tri(2-pyridyl)- 1,3,5-triazine)	[45]
Cupric ion reducing antioxidant capacity (CUPRAC) assay	Reduction of cupric ions to cuprous ions	[62]
Oxygen radical absorbance capacity (ORAC)(HAT)	Fluorescence generation	[63]
Thiobarbituric acid reactive substance (TBARS) assay	Fluorescence generation	[56]
Lipid peroxidation inhibition capacity (LPIC)	Fluorescence generation	[57]
Hydroxyl radical scavenging activity	Scavenging activity of antioxidants measured spectrophotometrically	[55]
NO free radical scavenging activity	Spectrophotometric method	[58]
Metal chelating activity	Decolorization of iron-ferrozine complex with fungal extracts	[64]

^a^ DPPH = 1,1-Diphenyl-2-picrylhydrazyl free radical, ^b^ ABTS = 2,2′-Azinobis (3-ethylbenzothiazoline-6-sulfonic acid)-diammonium salt.

## 3. Class of Marine Fungi Derived Antioxidant Molecules: Activities and Extraction Methodologies

Various classes of molecules of microbial origin shown to exert antioxidant and scavenging activities; they are not only represented by small metabolites such as phenolic compounds, anthraquinones, xanthones, carotenoids, indole derivatives and alkaloids, but also by polymers such as carbohydrates [65,66,67].

The extraction and purification strategies employed to afford the pure antioxidants, depend on the class of molecules and the type of cultivation (solid or liquid). Generally, several fungal metabolites can be found in both mycelium and culture broth. Small molecules extraction is almost always carried out by liquid-liquid extraction with organic solvents from the exhausted broth; frequently, ethyl acetate (EtOAC) is employed thanks to its immiscibility with the aqueous medium. More rarely adsorbent resins are used to absorb metabolites from the medium, with the aim to perform an elution with organic solvent in a second step [68]. Differently, when the fungal growth is conducted in solid media, different strategies are also possible, including serial partitioning with different solvents following a precise polarity gradient, including methanol (MeOH), acetone, ethanol (EtOH), n-butanol (n-BuOH), and water (H_2_O) [69,70]. Afterwards, crude extracts purification is pursued by different approaches, including silica gel, LH-20 and HPLC fractionation. Differently, polymers such as carbohydrates require different purification steps. 

In this paragraph we reviewed the different antioxidant metabolites isolated from marine fungi and the commonly used extraction strategies (Figure 1 and Figure 2 and Table 2).

### 3.1. Carbohydrates 

Carbohydrates are the most important building blocks of the life, consisting of carbon, hydrogen, and oxygen with the formula C_n_H_2n_O_n_. Based on the number of units, they are classified as monosaccharides, oligosaccharides that contain 3-10 monosaccharide, and polysaccharides (PSs) that consist of more than 10 monosaccharides.

PSs, have great diversity of structure and property based on the monomer’s composition, glycosidic linkages (β-1,3, β-1,6, and α-1,3) and grade of branching [71,72,73,74]. This variability gives them a wide range of applications in many fields of science and industry [75]; natural PSs are good candidates for therapeutic applications, since they show a variety of biological activities, such as immunomodulating and antitumoral [76]. In particular, while monosaccharides and oligosaccharides hardly exhibit antioxidants activities [77], PSs are reported to be effective radical scavengers and antioxidants [78].

Microbial-derived PSs are mainly divided in cell wall PSs, exopolysaccharides (EPSs), and endopolysaccharides (or cytoplasmatic PSs, CPSs), on the basis of their location in the cell [72]. 

According to the extensive in vitro antioxidant studies, fungal PSs, and in particular EPSs, are very effective antioxidants, but the mechanisms of action are still not clear [78], and need to be further confirmed; the iron chelating activity of EPSs could be involved, as iron is an important pro-oxidant [79]. Likewise, the factors that affect the antioxidant activity of EPSs are still not clear, but the presence of glucuronic acids (e.g., polygalacturonic acid, glucuronic acid, galacturonic acid) [80,81], or chemical modification (e.g., sulfatation) [82], are the most relevant.

Among them, EPSs show some advantages over the others, such as faster production and easier purification [83]. More than 50% of marine microbes are surrounded by EPS and, among them, bacteria and fungi are the most common [84]. Contrarily to cell wall or CPS, EPSs do not require organic solvents extraction [83,85]. Generally, the fermented broth of fungi is filtered and the resulting free-cell liquid is concentrated under vacuum at 40 °C and precipitated by threefold or fourfold volumes of ethanol 95% at 4 °C for 24h [86,87]. The obtained precipitate is dialyzed against distilled water for 48-72h, vacuum or frieze-dried, and the protein fraction is removed by Sevag method [88,89,90]. The crude EPS is fractionated by anion exchange chromatography and the obtained subfractions are dialyzed and further purified by gel filtration obtaining a pure polysaccharide (Figure 2) [91,92].

Recently an EPS, N1, composed by mannoglucogalactan with side chain of galactofuranose units was isolated by the deep-sea fungus *Aspergillus versicolor* N_2_bc, with 1.26 g/L of the crude PS yield. N1 exhibits a strong antioxidant activity in vitro how it is demonstrated by its half-maximal effective concentration values towards different radicals(EC_50_), equal to 0.97 mg/mL on DDPH radicals, 2.20 mg/mL on superoxide radicals, and 2.23 mg/mL on hydroxyl radicals [93]. 

Similarly, Chen et al. investigated the bioactivity of a novel EPS, composed by a mannan core with a galactoglucan chain, produced by an endogenous marine fungus *Alternaria* sp. The yield of the crude EPS was of 2.5 g/L, while the purified fraction, AS2-1, was about 65% of the former. AS2-1 was tested for its antioxidant activity by DPPH and hydroxyl radicals with EC_50_ values of 3.4 and 4.2 mg/mL, respectively. The cytotoxic effect of the purified EPS was evaluated on Hela (cervical cancer), HL-60 (human leukemia), and K562 (myelogenous leukemia) cell lines, with half-maximal inhibitory concentration (IC_50_) values of 0.167, 0.143, and 0.46 mg/mL, respectively [94].

A novel EPS, YSS, mainly composed by mannose and galactose was found in marine fungus *Aspergillus terreus* isolated from *Barracuda*. The yield of crude EPS was about 0.69 g/L, in which pure YSS represent about 45% of the total. This showed good scavenging capacity with the EC_50_ value of 2.8 mg/mL toward DPPH free radical and a lower scavenging ability on superoxide radicals than ascorbic acid, with EC_50_ value of 3.8 mg/mL [95]. 

The antioxidant activity of a novel EPS, AVP, produced by the coral-associated fungus *A. versicolor* LCJ-5-4 was also evaluated. AVP consists of mannoglucan and side chain of mannopryranose trisaccharides units. Herein, before the precipitation step, the fermentation broth was extracted three times with ethyl acetate to remove the liposoluble compounds. The resulting aqueous layer was further subjected to all the steps described above (Figure 2). The antioxidant activity was assessed by both DPPH and superoxide radicals scavenging ability with EC_50_ values of 2.05 and 4.0 mg/mL, respectively [96]. 

The antioxidant effect of the EPS fractions (ENP1 and ENP2) obtained from the marine fungus *Epicoccum nigrum* JJY40 was also studied, showing moderate antioxidant towards the DPPH, hydroxyl superoxide, and hydrogen radicals scavenging activities and lipid peroxidation inhibition. In particular, ENP2 and ENP1 showed EC_50_ values of 0.38 and 0.43 mg/mL, respectively. The authors suggested that the content of uronic acid in ENP2 could increase its antioxidant activity [81]. It is interesting to note that, to obtain the crude extract, before the dialysis step (Figure 2) the precipitate was washed three times with 95% ethanol, anhydrous ethanol, and acetone, respectively [81]. Moreover, in a previous work, three PSs, PS1-1, PS1-2 and PS2-1, with antioxidant activity were isolated from the marine fungus *Penicillium* sp. F23-2. These three EPSs are primarily consisted of mannose with variable amounts of glucose and galactose, while their glucuronic acid contents, molecular weights and glycosidic linkage pattern were different. PS1-1, PS1-2 and PS2-1 showed strong scavenging ability especially on superoxide and hydroxide radicals, with EC_50_ values in range of 0.36–1.14 and 0.18–1.13 mg/mL respectively, while EC_50_ values for DPPH and lipid peroxidation were in the range of 2.53–6.81 and 1.39–3.12, respectively. The effectiveness was in the decreasing order of PS2-1 > PS1-2 > PS1-1. The authors concluded that this evidence together with the differences in the molecular weight and glucuronic acid content of PS2-1, respectively lower and higher than other two EPSs, might explain the increased antioxidant activity [97]. 

Differently, CPSs extraction requires stronger methods than EPSs. In this context, Jaszek et al. described a very high ROS-scavenging potential of CPS isolated from *Cerrena unicolor*. The mycelia were collected from cultures by filtration, washed with distilled water, dried and weighed. Then the PSs were extracted with hot water (90 °C, 4h) 1:100 (*w*/*w*), cooled, and precipitated by fourfold volumes of cold ethanol overnight. The precipitate was collected by centrifugation and washed with ethanol. The antioxidative properties were assessed by three methods, chemiluminescence of luminol and ABTS and DPPH radicals scavenging, obtaining EC_50_ values of 0.183, 0.493 and > than 0.800 mg/mL, respectively [98].

The antioxidant properties of a sulphated derivate of a novel CPS, YCP, isolated from the mycelium of the marine filamentous fungus *Phoma herbarum* YS4108 were also investigated by Yang et al. [82]. The structure of the polymer consists of glucose and traces of glucuronic acid (1.5%). The crude polymer from *P. herbarum* YS4108 was obtained using different solvents and several steps, such as hot water (80 °C) extraction, filtration, deproteination of the filtrate through Sevag method, dialysis, ethanol overnight precipitation (4 °C), sequential washes of the collected precipitate with ethanol, acetone and ether, and in vacuo exsiccation. Finally, purified fractions were obtained by anion exchange and gel filtration chromatography. Sulfation of both crude extract and fractions was achieved using dry pyridine and chlorosulfonic acid to prepare the sulfating agent [82]. They concluded that sulfation of PS significantly increases its antioxidant capacity, such as superoxide and hydroxyl radicals scavenging activity, metal chelating action, lipid peroxidation and linoleic acid oxidation inhibition capability [82].

### 3.2. Phenolic Compounds

Phenolic compounds are known for their innate antioxidant activity, and several recent studies report a direct proportionality between phenolic compounds content in fruits and their antioxidant properties [99,100]. They are naturally produced by many plant species [101] and are involved in a complex signalling web between plants and the surrounding environment [102,103,104]. Their production is often linked to biotic and abiotic factors, such as stress and exposure to light [105,106]. Several phenolic metabolites are instead produced as antibacterial and/or antifungal agents, e.g., the stilbene pinosylvin [107]. All those which present at least on phenolic hydroxyl are classified as phenolic compounds [108] and are often grouped in phenols or polyphenols, hydroxybenzoic acid and hydroxycinnamic acid derivatives, flavonoids, stilbenes, some others are polymeric metabolites such as tannins and lignans [67,109]. 

However, plants are not the only organisms capable of producing metabolites displaying antioxidant features; many algae are able to produce several phenols and bromophenols with antioxidant activity [110,111], along with several fungi, many of which marine-derived are reported to biosynthesise phenolic derivatives and other classes of antioxidant metabolites [112,113]. Similarly to plants, fungal secondary metabolites expression is strongly regulated by external factors and stimuli, which often cannot be emulated during normal laboratory experiments. For this reason, most of their biosynthetic gene clusters (BCG) are silent under normal condition, thus singular strategies are often used to unlock their hidden potential [114,115,116]. Phenolic compounds extraction protocols depend on the sample source, and fungal derivatives are commonly obtained from both fungal exhausted broth and mycelium after separation by centrifugation. How is reported in Table 2, the extraction from the broth is commonly performed with EtOAc [117,118,119], which is the most commonly used solvent for liquid-.liquid extraction of small metabolites from microbial cultural broth. Moreover, given the extremely polar nature of this class of compounds, whenever possible, the use of polar solvents such as n-BuOH and MeOH allowed a better recovery of these polar molecules [112,120]. In fact, the mycelium is often treated with more polar organic solvents, centrifuged to remove the pellet, and eventually the pure compounds are afforded through ad hoc chromatographic purification of crude extracts [120].

Recently, nicotinamide and sodium butyrate, two histone deacetylases (HDACs) inhibitors, were used in the cultures of the marine-derived fungus *Penicillium brevicompactum* to stimulate the production of phenolic compounds, which was under-expressed in normal conditions. Extract purification gave nine phenolic compounds, among them syringic acid, acetosyringone, and sinapic acid showed a strong scavenging activity at the DPPH assay, with IC_50_ respectively of 20 ± 0.09, 25 ± 0.05 and 30 ± 0.08 μg/mL, together with a relevant antiproliferative activity towards human liver cancer cells [121]. 

Marine fungi isolated from ascidians (*Styela canopus*), seaweeds and sponges (*Hippospongia communis*) were subjected to three different in vitro tests to assess their antioxidant and anti-inflammatory capacities [117]. *Gymnascella dankaliensis, Nigrospora oryzae* and *Chaetomium globosum* extracts resulted particularly active at the DPPH, NO inhibition and TBARS assay, respectively. The evaluation of gallic acid concentration resulted to be higher in *Engyodontium album* and totally absent in *C. globosum.* The highest total phenolic content (TPC) was found in *G. dankaliensis,* while the highest total flavonoids content (TFC) was recorded for *E. album* [117].

The production of antioxidant secondary metabolites was also observed in the deep-see fungus *A. versicolor* [68,70,122]. The fungal grown was carried out on rice, and the fermented material was extracted utilizing EtOAc, n-BuOH, and H_2_O to afford the secondary metabolites. Coherently with their polarity, DPPH assay highlighted the presence of the antioxidant activity only in the n-butanol fraction. Once purified gave a total of 22 natural products belonging to phenolic compound and anthraquinone classes, among which six were new molecules. Antioxidant Capacity (TEAC), revealed that all the phenolic compounds displayed a scavenging activity close or higher than the positive control, with fumalic acid (5.24 ± 0.02), 1-methylpyrogallol (4.86 ± 0.05), cordyol C (5.40 ± 0.33) and lecanoric acid (5.06 ± 0.56) being the most actives [112]. Subsequently, 28 other phenolic compounds were isolated from the same fungus [68], many of which holding an anthraquinone core including aspergiols G-I, isolated for the first time together with 4-carbglyceryl-3,3’-dihydroxy-5,5’-dimethyldiphenyl ether and 2,4-dihydroxy-6-(4-methoxy-2-oxopentyl)-3-methylbenzaldehyde. At the DPPH assay, five metabolites gave a stronger antioxidant activity compared with L-ascorbic acid: 6-methylbenzene-1,2,4-triol (IC_50_ 35.08 μM), cordyol C (IC_50_ 31.16 μM) and sydowiol B-D (IC_50_ 21.22, 25.18, and 18.92 μM, respectively). These results revealed a direct proportionality between the scavenging activity and the number of the freely rotating phenolic hydroxyl groups, indicating the marine fungus *A. versicolor* as a very prolific strain for this purpose.

Methyl 4-(3,4-dihydroxybenzamido) butanoate, a new benzamide derivative with strong antioxidant activity was isolated from the algal-derived fungus *Aspergillus wentii* EN-48, together with three other phenolic derivatives. Noteworthy was the scavenging activity towards DPPH radical displayed by the new metabolite and 4-(3,4-dihydroxybenzamido)butanoic acid, resulting in 5.2 ± 0.3 and 9.6 ± 0.3 IC_50_ values, respectively, which were stronger than BHT (IC_50_ 36.9 ± 1.2) used as positive control [123].

Several Benzaldehyde derivatives from marine-derived fungi have also attracted interest for their scavenging properties. Wang et al. found and characterized for the first time chaetopyramin, a metabolite with scavenging properties obtained from the marine fungus *C. globosum* isolated from the red algae *Polysiphonia urceolata*. Chaetopyramin was obtained together with known derivatives isotetrahydroauroglaucin and 2-(2′,3-epoxy-1′,3′-heptadienyl)-6-hydroxy-5-(3-methyl-2-butenyl)benzaldehyde, showing IC_50_ values at the DPPH assay of 35, 26 and 88 μg/mL, respectively [120]. 

In this context, the marine-derived fungus *Microsporum* sp. was the source of two more benzaldehyde derivatives, flavoglaucin and isodihydroauroglaucin. These metabolites, known for their scavenging potential given by the presence of two phenolic hydroxyl groups, displayed a strong activity at the DPPH assay with IC_50_ values in order of 11.3 and 11.5 μg/mL, more potent than ascorbic acid (20 μg/mL) [120].

Abdel-Lateff et al. were able obtain from the marine fungus *Epicoccum* sp., associated with the algae *Fucus vesuculosus*, five natural products. Among them the new 4,5,6-trihydroxy-7-methylphtalide and (−)-(3R)-5-hydroxymellein displayed respectively, 95.2% and 22.4% of scavenging at 25 μg/mL by the DPPH assay, and 62.1% and 30.4% at 37 μg/mL by the TBARS assay [124]. 

Benzyl alcohols are among the simplest metabolites holding an antioxidant potential and are often isolate from several microorganisms; for instance, gentisyl alcohol and 3-chloro-4,5-dihydroxybenzyl alcohol were obtained from the algal-associated fungus *Aspergillus parasiticus*. The two alcohols were able to scavenge several free radicals with IC_50_ ≤ 11 mM [125].

Furthermore, several studies aimed to exploit the capacity of some marine fungal strain to biotransform natural products in more active molecules [126]. The marine fungus *Chrysosporium synchronum* was able to convert the previously isolated chlorogentisyl alcohol [127] into 1-O-(a-D-mannopyranosyl)chlorogentisyl alcohol, as the result of a two-step fermentation. This conferred to the glycosylated alcohol different phisyco-chemical features, but still holding a comparable antioxidant power with chlorogentisyl alcohol. They showed respectively IC_50_ values of 4.7 μM and 1 μM at the DPPH assay, both resulting stronger than L-ascorbic acid (20.0 μM) [119].

Four active hydroquinone derivatives were isolated from the marine fungus *Acremonium* sp., the new 7-isopropenylbicyclo (4.2.0)octa -1,3,5-triene-2,5-diol which presents an unusual condensed four-carbons ring, Gliomastin C, Gliomastin D, and F-11334A_1_. All the metabolites displayed good scavenging percentages at 25 μg/mL during the DPPH assay (respectively 85.5, 85.8, 72.9 and 90.2%), along with the property to inhibit peroxidation of linoleic acid at 37 μg/mL (35.5, 15.8, 9.2 and 16.6%) [128].

The hydroquinone farnesylhydroquinone was isolated together with its oxidized form, the sesquiterpene quinone, from the marine fungus *Penicillium* sp., and Farnesylhydroquinone (IC_50_ 12.5 μM) resulted a stronger DPPH radical scavenger compared with ascorbic acid (IC_50_ 22.5 μM) [129].

In addition, two new toluhydroquinone derivatives of which one brominated, were isolated from the algal-associated fungus *Dothideomycete* sp. together with two known hydroquinone derivatives. The new metabolites: 5-bromotoluhydroquinone and 4-O-methyltoluhydroquinone, together with the known toluhydroquinone and gentisyl alcohol resulted very active to the DPPH assay with IC_50_ values of 11.0, 17.0, 12.0, and 7.0 μM, respectively [118].

### 3.3. Carotenoids

Carotenoids are cluster of fat-soluble pigments primarily present in photosynthetic organisms and found produced also by many fungal species (especially pigmented yeasts) [66]. Carotenoids are biochemically terpenoids pigments synthesized from a 5-carbon precursor, isopentenyl pyrophosphate (IPP) from the parent compound mevalonate by isoprenoid pathway [130]. These compounds contain an aliphatic polyene chain comprising of eight isoprene units including light-absorbing conjugated double bonds, providing characteristics orange, yellow, red colors for the carotenoids [131]. Based on their structure, carotenoids are classified into two types namely carotenes (pure hydrocarbons) and xanthophylls (oxygenated carotenes) [132]. Carotenoids are potential antioxidants and possess protective effects on microorganisms against oxidative damage. Most common carotenoids produced from marine fungi include astaxanthin and β-carotene owing to their potential commercial viability [66].

Among the known carotenoids derived from marine fungi, Astaxanthin, a xanthophyll carotenoid is the strongest anti-oxidants due to its structure and better scavenging activity [65]. 

Astaxanthin is a lipophilic compound and extracted by solvents, acids, microwave coupled and enzymatic methods [132]. In *Phaffia rhodozyma,* red basidiomycetous yeast, astaxanthin is present in the cytoplasmic membrane with a rigid cell wall which makes it crucial for the extraction of the pigment. Sedmak et al. [133] utilized hot dimethyl-sulphoxide (DMSO) for the extraction of astaxanthin, but this method had a disadvantage for production of food/pharmaceutical grade astaxanthin due to the presence of DMSO residue in its crude form [133]. Storebakken et al. demonstrated enzymatic extraction method to be time-consuming and at the same time causing simultaneous degradation of astaxanthin [134]. Later, the astaxanthin extraction process by acidic method was developed, it overcame above limitations in extraction procedure [135]. Using lactic acid, 1294.7 µg/g yeast dry weight astaxanthin were extracted which were about 85.4% of total carotenoids amount. Emulsion-based systems are growing interests in encapsulating and delivering astaxanthin. β-cyclodextrin [136] and hydroxypropyl-β-CD [137] emulsions have found to be disperse astaxanthin. One such attempt was made by Villalobos-castillejos and his team where they produced a dispersible astaxanthin oleoresin from *P. rhodozyma*, at 65 °C for 24 min with a solvent: solid (ethyl acetate: yeast extract) ratio of 19:1, 6249 µg astaxanthin/g yeast with ethyl acetate as extraction solvent and polyethylene glycol as dispersive agent [138]. Recently, Chen and his team developed microencapsulated astaxanthin from *P. rhodozyma* cells. Physio-chemical methods including yeast cells disruption by glass beads, emulsifying the astaxanthin in aqueous phase by high shearing force with the help of gelatin and porous starch. The microencapsulation of astaxanthin was optimized with 1550 µg astaxanthin/g yeast was achieved [139]. He et al. devised optimum fermentation conditions for astaxanthin production using a medium containing glucose 8 g/L and peptone 8 g/L at 25 °C with pH of 5.5 from the marine yeast *Rhodotorula glutinis* YS-185, in which 2.67 μg/mL was achieved [140]. This strain has also been reported in production of lycopene, β-carotene, astaxanthin, torulene and torularhodin [141].

Among the uredininomycetous yeasts, β-carotene, torulene, torularhodin are very common types of carotenoids that includes the genera of *Rhodoturula, Rhodosporidium* and *Sporobolomyces.* β-carotene is the widely used commercial carotenoid type unlike torulene, torularhodin [142]. Kristian & Synnøve isolated a carotenoids-producing yeast strain from marine copepod *Calanus finmarchius,* which was identified as *Rhodosporidium babjevae* (Golubev). Carotenoids were extracted through the acid extraction method by employing HCl 1 M. At temperature 37 °C of growth, total carotenoid content was measured from 66 to 117 μg/g dry cell weight [143]. Interesting part of this work given to the identification of 3 class of carotenoids namely β-carotene, torulene and torularhodin. Zhao et al. isolated and identified carotenoid producing *Rhodotorula* sp., RY1801 a red pigment producing yeast. They optimized carotenoids production with an incubation period of 3 days, at 28 °C, pH 5.0, in a medium composed of 10 g/L glucose and 10 g/L yeast extract, a maximum concentration of 987 µg/L of total carotenoids was obtained [144]. *Rhodotorula mucilginosa,* isolated from sea weed *Macrocystis pyrifera* showed the production of carotenoids lycopene (38.4 ± 9.4%), β-carotene (21.8 ± 1.5%), astaxanthin (1.8 ± 0.3%) [145]. The optimum condition was found to be 25 °C for 7 days of yeast cultivation, the culture was then extracted with methanol/chloroform mixture (ratio of 2:1). A new red yeast *Sporobolomyces*
*ruberrimus* H110 was found to produce torularhodin, β-carotene of 2.70 mg/g and 0.10 mg/g dry weight of yeast, respectively, utilizing pure glycerol as the sole carbon source and 30 g/l media of ammonium sulphate as nitrogen source when grown at 23 °C and pH 6.0. The same yeast species improved the carotenoid production when raw glycerol was used as sole carbon source. The highest concentration of 0.51 g/L carotenoids was generated and torularhodin, torulene, ɤ-carotene and β-carotene were identified by mass spectrometric analysis, although individual quantitative production of carotenoids was not mentioned [146]. These studies depict the necessity for the advances in setting optimal conditions in extracting such bioactive compounds involving biochemical and analytical techniques when encountering novel microorganisms from marine environment. Neurosporaxanthin β-D-glucopyranoside, neurosporaxanthin β-carotene and γ-carotene with antioxidant activity were isolated from marine fungi *Fusarium*T-1 [147]. Only a smaller genus of marine fungi/yeast have been reported in production of these pigmented compounds carotenoids. Given the faster growth rates and easier cultivation of marine fungi, these organisms are off research interest in exploring the production of wide range of carotenoids. Conventional growth methods coupled with genetic engineering approach will allow in increased production of carotenoids. One such is already reported in *thrauschytrids* (fungi-like protists) where the production of carotenoid (astaxanthin) have seen nine-fold increase in comparison to normal method in *Aurantiochytrium* sp. SK4 [148]. Therefore, development of genetic tool and genome sequencing applied to marine yeasts/fungi will expand our knowledge of production of carotenoids, which will be useful in various field of biotechnology. 

### 3.4. Anthraquinones and Xanthones

Anthraquinones represent a large group of quinoid pigments with about 700 molecules described that differ in the nature and position of substituent groups [149], which are produced by plants, lichens, insect, bacteria and fungi [150,151,152]. Despite these molecules are mainly utilized as dyes in textile industry [153,154,155], anthraquinones show also a wide array of biological activities including anticancer, anti-inflammatory, antifungal, antibacterial, antiviral, and antioxidant [156,157,158,159,160,161]. 

Anthraquinones are widespread in fungi, as they were found in several genus including *Aspergillus* spp., *Eurotium* spp., *Fusarium* spp., *Dreschlera* spp., *Penicillium* spp., *Emericella purpurea*, *Culvularia lunata*, *Mycosphaerella rubella*, *Microsporum* sp. [149,162]. As in the case of other polyketide compounds, in fungi they are synthetized through the acetate-malonate pathway and are regulated by non-reducing polyketide synthases (NR-PKS′s) [149,163]. The involvement of both acetate and malonate is been studied through experiments using [^13−14^C-acetate] and [^13−14^C-malonate] [164,165,166,167]. 

Their basic structure is built from an anthracene ring with formula C_14_H_8_O_2_ and ketone groups on the central ring in position C-9 and C-10 as basic core [152]. There are eight possible hydrogens that can be replaced with other different functional groups, such as -OH, -CH3, -OCH3, -CH2OH, -CHO, -COOH, or more complex groups, and can be either in free form, as glycosides, or other complexes attached through an O or C bond in the side chain, which increase their water solubility [149]. When hydrogens are substituted only by hydroxyl groups, the molecule is called hydroxyanthraquinone (HAQN) [151].

Anthraquinones can be present either in the mycelium of fungi or in the fermentation broth, for this reason extraction techniques are based on the exhausted culture broth and mycelia cake or the whole fermentation broth [70,168]. Depending on their nature, different solvents can be employed to extract anthraquinones. In fact, usually these compounds are extracted with low-polarity solvents such as chloroform, dichloromethane, and ethyl acetate, while for glycosylated anthraquinones solvents with high polarity are utilized, such as methanol, ethanol, and water [150].

Anthraquinones and their derivatives showed clear antioxidant activity [169,170,171,172,173], because of the quinoid structure that allow them to participate in redox reactions [149]. This is not infrequent in aromatic compounds, but as for vitamin C, carotenoids or flavonoids, at higher concentrations or in certain conditions, these compounds may present toxic prooxidant activity [162,174]. The toxic and prooxidant activity of some anthraquinones could be related to the formation of semiquinone radicals that lead to the formation of singlet oxygen [149,175,176].

Xanthones are another large class of natural polyphenolic compounds, with more than 200 molecules described from plants, lichens, fungi, and bacteria with molecular formula of C_13_H_8_O_2_ [177,178]. They are tricyclic compounds with a symmetric structure, derived from dibenzo-γ-pirone [179,180] and are known to have a multiplicity of biological activities, such as antioxidant [181], antiproliferative [182], antimicrobial [183], antitubercular [184], antitumoral [185], and antiviral [186] because of their different interaction with a range of molecular target [187]. Each molecule has a three-ring skeleton and differ from each other by the position on the central ring of different substituents [188]. Based on their structures, xanthone derivates can be divided in six main groups, simple oxygenated xanthones, xanthone glycosides, prenylated xanthones, xanthonolignoids, bisxanthones, and various xanthones [189]. The oxygenated xanthones are further subdivided according to the degree of oxygenation in non, mono-, di-, tri-, tetra-, penta- and hexa-oxygenated substances [190], xanthones glycosides are divided in C- or O-glycosides [191]. Prenylated xanthones are produced only by plants, while bisxanthones are produced also by lichens and fungi, and natural xanthonolignoids are rare and only few are known [178]. 

In fungi, xanthones are mainly synthetized by the acetate pathway through the oxidative cleavage of an anthraquinone to give a benzophenone. At this point, depending on the producing organism, xanthones can be formed directly from the benzophenone or through a tetrahydroxanthone intermediate [179,192].

However, the antioxidant properties of these molecules depend from their structure and the methods of assessment [161]. Wu et al. investigated the antioxidant features of five anthraquinones, aspergiol A, aspergiol B, both described for the first time, averythrin, averantin, and methylaverantin, produced by the deep-sea fungus *A. versicolor*, through ABTS assay [112]. The new compounds showed activity significantly higher than Trolox, while the others comparable with it. Recently, from the same strain eight anthraquinones, versicolorin B, UCT1072M1, averantin, methylaverantin, averythrin, averufanin, averufine, and nidurufin, and four xanthones, oxisterigmatocystin D (isolated for the first time), oxisterigmatocystin C, sterigmatocystine, and dihydrosterigmatocystine, were obtained in the same conditions and with the same procedures described before, with the only difference consisting in the fractionation and purification of the ethyl acetate extract. Four of the obtained molecules showed activity equivalent to the Trolox [70].

Huang et al. isolated nine anthraquinones, aspergilol G, aspergilol H, aspergilol I, for the first time, aspergilol A, aspergilol B, SC3-22-3, coccoquinone A, averufin, methylaverantin, and versiconol, and a xanthone with antioxidant activity produced by the deep-sea fungus *A. versicolor*, SCSIO 41502 [68].

Du et al. isolated from *Aspergillus europaeus* WZXY-SX-4-1, twenty polyketide derivates of which six xanthones, four anthraquinones, and two seco-anthraquinones [189]. Among the xanthones, euroxanthone A and euroxanthone B were isolated for the first time, while the others correspond to the known calyxanthone, yicathin A, yicathin B, and yicathin C. Among the anthraquinones, the new 1-O-demethylvariecolorquinone A, together with the already known variecolorquinone A, dermolutein, and methylemodin were isolated, while for the seco-anthraquinones were isolated 1-methoxy-14-dehydroxywentiquinone C and wentiquinone C. The structure of the wentiquinone C recently revised to be 8-hydroxy-6-(hydroxymethyl)-3-methoxy-9-oxo-9H-xanthene-1-carboxylic acid by Li et al., assigning it to the xanthones family instead of anthrones [193]. Following the OSMAC (One Strain MAny Compounds) method, the authors found that the best condition for the expression of the antioxidant activity of the fungus was culturing it on solid media (salty rice) for 21 days at 25 °C, and to extract it with EtOAc. The scavenging activity of these fractions was investigated through DPPH assay, resulting that the most active was the fraction 2 with IC_50_ of 10 µg/mL, while the others showed IC_50_ > 100 µg/mL. Fraction 2 and 3 were further fractionated by RP column chromatography eluting with MeOH-H_2_O to obtain all the isolated compounds. Finally, the authors reported the compounds with the higher scavenging activity against DPPH in which were present variecolorquinone A and yicathin C with IC_50_ of 13.2 and 17.5 µg/mL, respectively, while Trolox showed IC_50_ of 5.4 µg/mL.

Despite these molecules are produced from both free and endophytic marine fungi, little is reported from the former and, to the best of our knowledge, this is the first review on the antioxidant activity of anthraquinones and xanthones from non-endophytic marine fungi.

### 3.5. Other Molecules 

Recently, several new indole derivatives have attracted attention thanks to their biological activities [193,194,195]. In this respect, tryptophol [196] was obtained together with two new derivatives: 2-(1*H*-indol-3-yl)ethyl 2-hydroxypropanoate and 2-(1*H*-indol-3-yl)ethyl 5-hydroxypentanoate, from the marine yeast *Pichia membranifaciens*. The DPPH assay highlighted significant scavenging percentages for the three metabolites at 300 μM (respectively 52, 33 and 47%) [197].

Terpenes and terpenoids are often obtained from marine endophytic fungus, isolated from superior organisms, including algae, sponges, tunicate [27,198]. Recently, the algiculous *Aspergillus* sp. SpD081030G1f1 was source of bioactive terpene analogues, such as the new JBIR-81 and JBIR-82 together with the known metabolite terpeptin. They were found to exert protective activity against L-glutamate toxicity in N18-RE-105 cells, with EC_50_ values were 0.7, 1.5 and 0.9 µM, respectively [69]. 

2-Hydroxycircumdatin C, a new benzodiazepine alkaloid was derived from the brown algae endophytic fungi *Aspergillus ochraceus,* most likely due to the presence of a catechol moiety, displayed a relevant DPPH radical-scavenging activity (IC_50_ 9.9 μM), about 9-folds higher compared to BHT [199].

Li et al. were able to isolate Golmaenone, a new diketopiperazine alkaloid together with the known derivative neochinulin A, from the marine fungus *Aspergillus* sp. Both metabolites showed similar scavenging activity to the ascorbic acid (IC_50_ 20 µM), along with Ultra Violet-A (UV-A) protecting activity [200].

Contextually, the algicolous fungus *Pseudallescheria* sp. was able to produce several alkaloids, in particular the dioxipiperazine gliotoxin demonstrated to be a stronger scavenger of the DPPH radical (IC_50_ value of 5.2 mM) compared to ascorbic acid [201].

Furthermore, parasitenone, a new epoxycyclohexenone obtained from *A. parasiticus*, resulted to be mildly active as scavenger towards DPPH and ONOO^-^ radicals (IC_50_ 57.0 and 52.6 mM, respectively) [125].

The antioxidant metabolite 5-(hydroxymethyl)-2-furanocarboxylic acid was isolated from the marine fungus *Wardomyces anomalus*, together with other xanthone congeners (discussed in the Section 3.4), and it displayed activity both as scavenger towards DPPH radical (30.7% at 25.0 mg/mL) and at the TBARS assay (2.6% at 7.4 mg/mL) [187].

The marine fungus *Acremonium strictum* was source of Acremostrictin, a secondary metabolite with the ability to scavenge DPPH radical (IC_50_ of 2.1 mM), besides it was also able to inhibit the H_2_0_2_ mediated death of human HaCaT cells [202].

Marine fungi *Phaeotheca triangularis*, *Trimmatostroma salinum*, *Hortaea werneckii*, *Aureobasidium pullulans* and *Cryptococcus liquefaciens* have been found to produce mycosporine-like amminoacids (MAA) such as mycosporine–glutaminol–glucoside and mycosporine–glutamicol–glucoside, that can absorb UV in the range of 310-320 nm [203]. MAAs may protect the skin against UV radiation; they also exhibit a high anti-oxidant activity, scavenging superoxide anions, and inhibiting lipid peroxidation [204,205,206]. The properties of MAAs as UV screens and ROS scavengers suggest that they could be used in sunscreen products [207].

**Table 2 antioxidants-09-01183-t002:** Summary of all the classes metabolites with antioxidant activity produced by marine fungi together with the adopted extraction methodologies.

Source	Strain	Antioxidant Molecule/s	Extraction Methodologies ^a^	Ref.
**Phenolic Compounds**				
*Callyspongia siphonella* (Sponge)	*Penicillium brevicompactum*	syringic acid, acetosyringone, sinapic acid	E.b. extraction: EtOAc	[121]
*Hippospongia communis* (Sponge)	Gymnascella dankaliensis, Nigrospora oryzae, Chaetomium globosum, *Engyodontium album*	Crude extracts	Mycelium extraction: EtOAc	[117]
Deep-sea sediments (depth of 3002 m)	*Aspergillus versicolor*	fumalic acid, 1-methylpyrogallol, cordyol C, lecanoric acid	Solid culture serial extractions: EtOAc, n-BuOH (active fraction) and H_2_O	[112]
Deep-sea sediments (depth of 2300-2400 m)	*Aspergillus versicolor SCSIO 41502*	6-methylbenzene-1,2,4-triol, cordyol C, sydowiol B-D	E.b. extraction: XAD-16 resin and elution with EtOH. Mycelium extraction: acetone. The two extracts were combined together.	[68]
*Sargassum* sp. (Brown algae)	*Aspergillus wentii* EN-48	4-(3,4-dihydroxybenzamido) butanoate	E.b. extraction: EtOAc. Mycelium extraction: acetone.	[123]
*Polysiphonia urceolata* (red algae)	*Chaetomium globosum*	Chaetopyramin, isotetrahydroauroglaucin, 2-(2′,3-epoxy-1′,3′-heptadienyl)-6-hydroxy-5-(3-methyl-2-butenyl)benzaldehyde	E.b. and mycelium homogenized extraction: MeOH and EtOAc.	[120]
*Lomentaria catenate* (red algae)	*Microsporum* sp.	flavoglaucin and isodihydroauroglaucin	E.b. extraction: EtOAc. Mycelium extraction: CH_2_Cl_2_:MeOH 1:1. The two extracts were combined together.	[120]
*Fucus vesuculosus* (brown algae)	*Epicoccum* sp.	4,5,6-trihydroxy-7-methylphtalide, (-)-(3R)-5-hydroxymellein	E.b. and mycelium homogenized extraction: EtOAc.	[124]
*Carpopeltis cornea* (red algae)	*Aspergillus parasiticus*	gentisyl alcohol, 3-chloro-4,5-dihydroxybenzyl alcohol	E.b. extraction: EtOAc. Mycelium extraction: CH_2_Cl_2_:MeOH 1:1. The two extracts were combined together.	[125]
*Sargassum ringgoldium* (brown algae)	*Chrysosporium synchronum*	1-O-(a-D-mannopyranosyl)chlorogentisyl alcohol from chlorogentisyl alcohol	E.b. extraction: EtOAc.	[119]
Algae (species not specified)	*Acremonium* sp.	7-isopropenylbicyclo[4.2.0] octa-1,3,5-triene-2,5-diol, Gliomastin C, Gliomastin D, F-11334A_1_	E.b. and mycelium homogenized extraction: EtOAc.	[128]
Marine sediment	*Penicillium* sp.	farnesylhydroquinone	Mycelium extraction: CH_2_Cl_2_:MeOH 1:1.	[129]
*Chondria crassicualis* (red algae)	*Dothideomycete* sp.	5-bromotoluhydroquinone, 4-O-methyltoluhydroquinone, toluhydroquinone, gentisyl alcohol	E.b. extraction: EtOAc.	[118]
**Anthraquinones and Xanthones**				
Deep-sea sediments (depth of 3002 m)	*Aspergillus versicolor* (A-21-2-7)	aspergiol A, aspergiol B, averythrin, averantin, and methylaverantin	Solid culture serial extractions: EtOAc, n-BuOH (active fraction) and H_2_O.	[112]
Deep-sea sediments (depth of 3002 m)	*Aspergillus versicolor* (A-21-2-7)	versicolorin B, UCT1072M1, averantin, methylaverantin, averythrin, averufanin, averufine, nidurufin, oxisterigmatocystin D, oxisterigmatocystin C, sterigmatocystine, and dihydrosterigmatocystine	Solid culture serial extractions: EtOAc, n-BuOH (active fraction) and H_2_O.	[70]
Deep-sea sediments (depth of 2300-2400 m)	*Aspergillus versicolor* SCSIO 41502	aspergilol G, aspergilol H, aspergilol I, aspergilol A, aspergilol B, SC3-22-3, coccoquinone A, averufin, methylaverantin, and versiconol	E.b. extraction: XAD-16 resin and elution with EtOH. Mycelium extraction: acetone. The two extracts were combined together.	[68]
*Xestospongia testudinaria* (sponge)	*Aspergillus europaeus* WZXY-SX-4-1	euroxanthone A and euroxanthone B, calyxanthone, yicathin A, yicathin B, yicathin C, new 1-O-demethylvariecolorquinone A, variecolorquinone A, dermolutein, methylemodin, 1-methoxy-14-dehydroxywentiquinone C and wentiquinone C	Solid culture extraction: EtOAc.	[189]
**Carotenoids**				
Marine environement	*Phaffia rhodozyma*	Astaxanthin	The mycelium was treated with Lactic acid 5.5 mol/L at 30 °C, then extracted with a EtOH.	[135]
Marine environement	*Phaffia rhodozyma*	Astaxanthin	Mycelium extraction: n-hexane:EtOAc (1:1)	[208]
Marine environment	*Rhodotorula glutinis* YS-185	Astaxanthin	Mycelium extraction: EtOAc.	[140]
*Calanus finmarchius* (copepods)	*Rhodosporidium babjevae* (Golubev)	Torularhodin, Torulene, β -carotene.	Cell suspensions added to 10% 1 M KOH in MeOH and neutralized with 1M HCl at 37 °C and extracted into ether	[143]
*Deep sea sediments*	*Sporobolomyces ruberrimus*	Carotenoids (torularhodin, torulene, β-carotene and ɤ-carotene)	Mycelium extraction: EtOH.	[146]
*Macrocystis pyrifera* (large brown algae)	*Rhodotorula mucilginosa*	Carotenoids (lycopene, β -carotene, astaxanthin)	Mycelium extraction: MeOH:CHCl_3_ (2:1).	[145]
*Marine environment*	*Rhodotorula* sp. RY1801	Carotenoids	Mycelium extraction: DMSO:Acetone (2:5)	[144]
*Tanegashima, Kagoshima Prefecture, Japan*	*Fusarium Strain T-1*	neurosporaxanthin β -D-glucopyranoside, neurosporax- anthin, β -carotene, γ-carotene, and torulene	Mycelium extraction: acetone	[147].
**Other Molecules**				
*Halichondria okadai* (sponge)	*Pichia membranifaciens*	Tryptophol, 2-(1*H*-indol-3-yl)ethyl 2-hydroxypropanoate, 2-(1*H*-indol-3-yl)ethyl 5-hydroxypentanoate	E.b. (pH 3.0) extraction: EtOAc. Then the extract was dried over Na_2_SO_4_ and filtered	[197]
*Sargassum* sp. (brown algae)	*Aspergillus* sp. SpD081030G1f1	JBIR-81, JBIR-82, terpeptin	Solid culture extraction: Acetone:H_2_O 8:2	[69]
*Sargassum kjellmanianum* (brown algae)	*Aspergillus ochraceus*	2-Hydroxycircumdatin C	E.b. and mycelium homogenized extraction: EtOAc and MeOH.	[199]
*Lomentaria catenata*	*Aspergillus* sp.	Golmaenone, neochinulin A	E.b. extraction: EtOAc.	[200]
*Agarum cribrosum* (brown-algae)	*Pseudallescheria* sp.	gliotoxin	E.b. extraction: EtOAc.	[201]
*Carpopeltis cornea* (red algae)	*Aspergillus parasiticus*	parasitenone	E.b. extraction: EtOAc. Mycelium extraction: CH_2_Cl_2_:MeOH 1:1 from mycelium, then combined together	[125]
*Enteromorpha* sp. (green algae)	*Wardomyces anomalus*	5-(hydroxymethyl)-2-furanocarboxylic acid	E.b. and mycelium homogenized extraction: EtOAc.	[187]
Sponge (species not specified)	*Acremonium strictum*	Acremostrictin	E.b. extraction: EtOAc.	[202]
Marine source (not specified)	*Phaeotheca triangularis*, *Trimmatostroma salinum*, *Hortaea werneckii*, *Aureobasidium pullulans* and *Cryptococcus liquefaciens*	mycosporine–glutaminol–glucoside, mycosporine–glutamicol–glucoside	The Lyophilized mycelium is pulverized under liquid nitrogen and extracted with 0.2% aqueous acetic acid supplemented with 0.5% MeOH (*v*/*v*) for 12h at 4 °C. [209]	[203]

^a^ E.b = Exhaust broth.

## 4. Trends and Perspective on the Market for Marine Fungal Antioxidants

Antioxidants are substances that, at low concentrations, significantly inhibit or delay the oxidation of biomolecules such as lipids, proteins, and DNA [210]. By virtue of this property, antioxidants find applications in different industrial fields.

In particular, antioxidants have been widely used in the food industry for decades, as a food additive, preventing lipids from oxidative degradation, or packaging [211]. Although antioxidants are naturally present as food components (e.g., tocopherol and ascorbic acid), the endogenous antioxidants are often, completely or partially lost during processing or storage, thus requiring their addition from exogenous sources [212]. In addition, these molecules are also used as feed additives in animal food. Moreover, they are also added in cosmetic and pharmaceutical formulations to prevent the oxidative degradation of ingredients and enhance the stability of therapeutic agents [213]. Despite the high demand for natural antioxidants to prevent diseases caused by oxidative stress, the complete process of bringing a new product to market, defined as New Product Development, is particularly long and complex for pharmaceuticals [214]. The entire process can take more than a decade, with a high rate of unsuccessful. The compounds have to successfully pass four phases of Clinical Trials before considered suitable for approval by the national regulatory authority for use in the general population. Therefore, it is clear that the time-to-market can be very long in comparison with other industrial fields, like food and cosmeceutical industries. These molecules play an additional important role in skin-aging products, together with photo-protective molecules, protecting the skin from oxidation induced by ROS and preventing the consequent visible signals due to the degradation of the extracellular matrix in both the epidermal and dermal layer [215].

The field of application of antioxidants is not limited to food, cosmeceutical, and pharmaceutical industry, in fact, antioxidants are frequently added to industrial products. They are commonly used as stabilizers in fuels and lubricants to prevent oxidation, and in gasoline to prevent the polymerization that leads to the formation of engine-fouling residues [216]. Moreover, they are widely used to prevent the oxidative degradation of polymers such as rubbers, plastics and adhesives that causes a loss of strength and flexibility in these materials [217].

At present, most of the permitted antioxidants authorised to be used in foods as well as in cosmetic and cosmeceutical products are artificial molecules, such as butylated hydroxianisole (BHA), butylated hydroxitoluene (BHT), propyl gallate (PG), and tertbutylhydroquinone (TBHQ) [218]. However, there is an increasing interest in seeking natural antioxidants due to the potential health hazard posed by synthetic antioxidants at high concentrations and during long-term intake [219]; carcinogenic effects of BHA and BHT in experimental animals have been reported [220,221,222,223].

In this perspective, marine fungi represent a promising source of antioxidant molecules. It is interesting to note that these molecules can be extracted at low cost by using waste materials as substrate for the fermentation of source organisms. Moreover, abundant fungal biomasses needed for the extraction of a large amount of bioactive compounds can be obtained by Solid-State Fermentation (SSF), which has proven to be an effective method for the bioconversion of different low-cost industrial by-product into valuable products, by cutting fermentation and bioactive compounds production costs [224]. In addition, the use of these residues as carbon sources through SSF provides an important way to avoid environmental problems caused by their disposal, being an economical and eco-friendly solution for countries with abundance of these materials.

However, so far, the use of fungal-derived antioxidant molecules in these applications is limited to the production of feedstuff [225,226] (see Section 4.1). On the contrary, there are many reports available for other marine microorganisms, mostly in cosmeceutical field. In this section, some of these examples are reported as references for the development of novel potential fungal antioxidants. Moreover, we focus our attention on regulatory issues that limit their use in food industry for human use.

### 4.1. Food Industry

Despite the strong interest in the use of alternatives to synthetic antioxidants, very few antioxidants have reached the market due to problems associated with regulatory issues.

The use of antioxidants for food application is governed by regulatory laws of the individual country or by internal standards. Long term toxicological studies of possible mutagenic, teratogenic and carcinogenic effects, using animal models, are crucial in determining the safety of an antioxidant and in determining the allowable daily intake (ADI) levels [218]. An antioxidant should have two conditions to be considered as safe: (i) its median lethal dose (LD_50_) must not be less than 1000 mg/kg body weight, and (ii) it should not have any significant effect on the growth of the experimental animal in long-term studies at a level 100 times greater than that proposed for human consumption [227]. Antioxidants are extensively tested for the absence of toxic effects in themselves, in their oxidized forms and in their reaction products with food constituents [218]. In addition to safety requirements, antioxidants to be used in food applications should satisfy several other requirements in order to be added to foods without affecting organoleptic and nutritional properties, and without being affected by food processing conditions like sterilization, boiling and evaporation. The antioxidant should be fat-soluble, stable to pH change and heat processing, effective at low concentrations and for at least 1 year at temperature 25 °C and 30 °C; it should not change flavours, odours and colours of food [218,228]. Some synthetic antioxidants have limitations to be used in food requiring severe thermal treatment, for example the gallates decompose when heated above 148 °C, and BHA and BHT are more susceptible to steam distillation [229,230]. Moreover, the use of synthetic antioxidants is limited for various reasons which include the polarity and size of the molecules [231].

Considering adverse effects of synthetic antioxidants at high concentrations and low thermal stability of them in heat processing and frying of food products, it seems logical to substitute synthetic antioxidants with alternative antioxidants from natural sources. Antioxidants molecules present in biological material have attracted considerable interest because of their presumed safety and potential therapeutic effects [232].

The most common sources of natural antioxidants currently exploited for food applications include herbs and spices (rosemary, oregano, thyme, tea, turmeric etc.), fruits (apple, strawberry, blueberry, cranberry, grape etc.) and vegetables (beetroot, carrot, kale, spinach, broccoli etc.). Other sources of natural antioxidants include agricultural and processing by-products (e.g., olive leaves) and antioxidants from animal sources (e.g., protein isolates) [233,234]. The antioxidant molecules extracted from plant and fruits currently exploited as food additives are mostly phenolic compounds [235].

In this contest, the astaxanthin production from natural sources has become a very interesting successful activities in biotechnology, as it is used in several commercial applications in the market.

The main astaxanthin sources are e the microalgae *Haematococcus pluvialis*, the yeast *Phaffia rhodozyma* and synthetic astaxanthin, from this sources a lot of patent application have been set, and plenty of products have been commercialized which have been carefully reported [66,225,236].

Even if yeasts and bacteria produce lower astaxanthin amounts compared with algae, they grow faster, and presents easier fermentation protocols, thereby they are promising candidate for biotechnological uses [66].

In particular, the marine yeast derived astaxanthin is actually used as nutritious feedstuff in animal diets, including fishes (salmons and trouts), crustaceans and poultry [225,226]. Astaxanthin commercial scale production is obtained by a *Phaffia rhodozyma* mutate strain and its commercialized under the products Ecotone^®^ and AstaXin^®^.

Even if there is no other evidence of marine fungi-derived antioxidant molecules exploited for food application, as reviewed above, different marine fungi are also prolific sources of phenolic compounds [112,118,128] that could find potential applications as antioxidant food additives, following a more in-depth study aimed at verifying the satisfaction of the requirements given by regulatory laws.

Bioactive phenolic compounds obtained by SSF using agro-industrial residues have been review by Martins et al. [237]. The fungal SSF constitute an advantage which makes marine fungi potential candidates for the development of new antioxidant food additives. An important limiting factor which makes the utilization of natural antioxidants as food preservatives, in fact, is the production cost. Natural plant antioxidants are typically more expensive than synthetic antioxidants because they must be extracted and purified from botanical sources, often in large quantities.

Another interesting possible application of fungal-derived antioxidant molecules in the food industry, could be their use in antioxidant active packaging. The demand for antioxidant active packaging is increasing due to their unquestionable advantages compared with the addition of antioxidants directly to the food, which include the lower amount of active compounds required, an activity focused at the surface of the food, extended antioxidant effect due to controlled migration from the film to the food matrix, simplification of industrial processing of the food (no need of additional steps such as mixing, immersion or spraying to include the active substances) [238]. Moreover, the accumulation of synthetic plastics, mainly from food packaging, is causing serious environmental problems, leading to a high demand for the development of biodegradable films and coatings. Microbial polysaccharides, due to their low cost, have emerged as new and industrially important biopolymers competing with natural gums obtained from marine algae and higher plants, for the development of biodegradable, eco-friendly and non-toxic food packaging films, among other application [239]. Marine fungi, as described in this review, have shown the ability to produce plenty of polysaccharides with antioxidant properties. Therefore, these polymers represent promising candidate for the development of biodegradable active packaging material.

### 4.2. Cosmetic and Cosmeceutical Industries

The use of marine ingredients in cosmetic and cosmeceuticals is not a new concept, and the cosmetic industry is constantly looking for innovative solutions. Selected marine-derived actives have been introduced in the formulations of prestigious skin care products, including Elemis (The Steiner Group, London, UK), La Prairie (Beiersdorf, Montreux, Switzerland), Crème de la Mer (Estée Lauder, New York, NY, USA), Blue Therapy (Biotherm, Tours, France), and many others [240].

The list of approved marine ingredients as part personal care products formulation include compounds and extracts obtained from macro algae, sponges, corals, jellyfish, crustacean and fish [241].

However, the popularity of marine ingredients is leading to concerns about sustainability and toxicity (Organic Monitor Website). For marine ingredients harvested from the ocean, there may be issues with sustainability since require large-scale sourcing with consequent negative impact on the surrounding environment and concern about heavy metals. For marine ingredients cultivated and grown in a carefully monitored environment, instead, both concerns can be lessened considerably. For this reason, microorganisms, including fungi, are considered the excellent sustainable source for the research of new molecules. Ingredients from this source can be developed cost-effectively at large-scale based on traditional fermentation technologies [242], with minimal environmental impact associated with their initial collection (unlike wild seaweed or some fish-derived ingredients). Moreover, as they can be grown in conventional fermentation facilities, manufacturing is readily scalable and production can be carried out near to the end-manufacturer, minimizing the carbon footprint associated with transportation.

Among microorganisms, microalgae-derived extracts have a long tradition in cosmetic industry, being incorporated in many face and skin care products (e.g., anti-aging cream, refreshing or regenerating care products, emollient and anti-irritant in peelers), sun protection and hair care products [243]. Some examples of cosmetic products containing microalgae-derived extracts are: Depollutine^®^ and Grevilline^®^, containing peptidic extracts from *Phaeodactylum tricornutum* and *Skeletonema costatum*, respectively; Porphyraline^®^, containing an extract from *Porphyridium cruentum**;* Dermochlorella DG^®^, containing the aminoacids and oligopeptides-rich extract from *Chlorella vulgaris*; XCELL-30^®^, containing an extract rich in potassium and 3-dimethylsulfopropionate (DMSP) from *Imperata cylindrica*; Alguronic Acid^®^, containing a microalgal oil; Alguard^®^, containing a polysaccharide solution from *Porphyridium* sp.

Moreover, marine bacteria-derived compounds approved as ingredients for cosmetic formulations include in particular polysaccharides. The exopolysaccharide named deepsane, produced and secreted by the strain *Alteromonas macleodii subsp. fijiensis biovar deepsane* [244] is commercialised under the name of Abyssine^®^ (patent PCT 94907582-4) for soothing and reducing irritation of sensitive skin against chemical, mechanical and ultraviolet B (UVB) aggression. Another example of an exopolysaccharide with tremendous market impact is SeaCode^®^ launched by LIPOTEC, a mixture of extracellular glycoproteins (GPs) and other glucidic exopolymers produced by biotechnological fermentation of a *Pseudoalteromonas* sp. isolated in the intertidal coasts of Antarctic waters.

RefirMAR^®^ by BIOALVO is an intracellular extract produced by biotechnological fermentation of a *Pseudoalteromonas* sp. strain isolated from the extreme environment of the Rainbow vent [245] and used as active ingredient–as potent hydrating, anti-wrinkle and expression lines attenuator –in personal care products [240].

Despite there is no evidence of approved marine fungi-derived antioxidants as ingredients of cosmetic formulations, the recent research progresses on the extraction methodologies, and the continuous interest of beauty industry towards innovative and new ingredients, make marine fungal antioxidants and photo-protective compounds (e.g., MAA and carotenoids) promising candidates for the development and their inclusion in cosmetic end-products [15].

## 5. Conclusions

The marine environment is gaining attention as a biotechnological resource. In the last two decades, the number of discovered bioactive marine natural products has progressively increased. Among marine organisms, fungi are considered a promising source, thanks to the high amount of Biosynthetic Gene Clusters (BGC) correlated to the production of secondary metabolites. In particular, a wide range of marine-fungal antioxidants has been reported, although just a few are employed in end products. However, recent reports on the use of other marine microorganisms-derived products, mainly in cosmetics and feed food, are opening the chance to draw from marine fungal metabolites in this as in other fields. The use of marine fungi-derived products would present several advantages compared to synthetic antioxidants and to those derived from not renewable sources such as plants and algae, in term of reduction of production costs and sustainability of the process. In fact, fungal fermentation is an easy-scalable process and fungi can be subjected to genetic manipulation. In addition, the ongoing efforts in the optimization of the extraction processes can be very helpful.

In conclusion, the great number of molecules with promising antioxidant activity reported in this work can find application in several fields, such as food, cosmeceuticals, nutraceutical, and pharmaceuticals (e.g., polysaccharides, MAA, astaxanthin and other carotenoids).

Further researches on the still poor explored field of marine fungi antioxidants can further enrich the plethora of active metabolites already reported. In addition, studies targeted at the development of novel genetic and metabolic tools aimed to their overproduction, can strongly enrich the market with novel marine-fungal antioxidants.

## Figures and Tables

**Figure 1 antioxidants-09-01183-f001:**
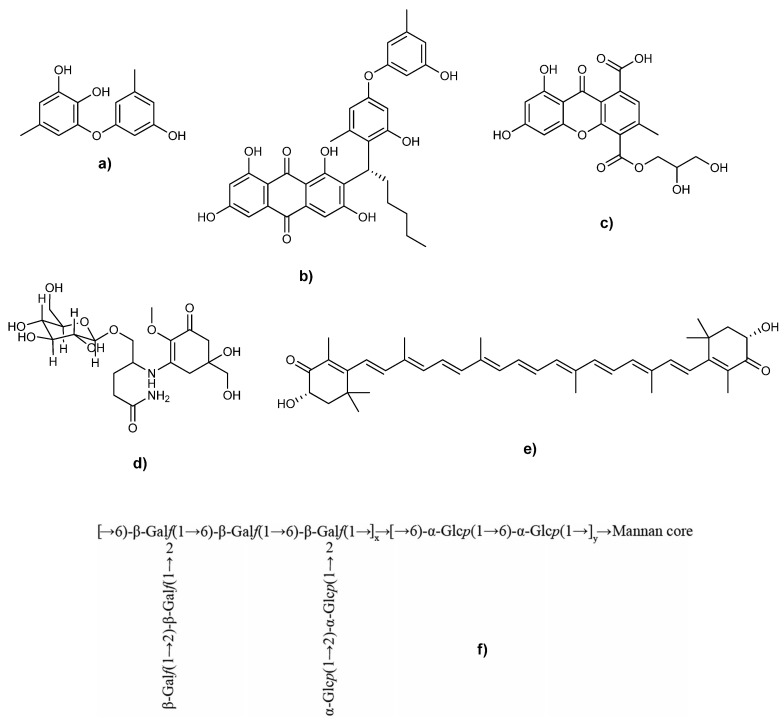
Some marine fungal metabolites with antioxidant properties in rappresentance of various chemical classes, respectively: (**a**) cordyol (phenolic compounds) C, (**b**) aspergilol (anthraquinones) A, (**c**) euroxanthone (xanthones), (**d**) mycosporine-glutaminol-glucoside (amino acids derivatives), (**e**) astaxanthin (carotenoids), (**f**) AS2-1 (carbohydrates).

**Figure 2 antioxidants-09-01183-f002:**
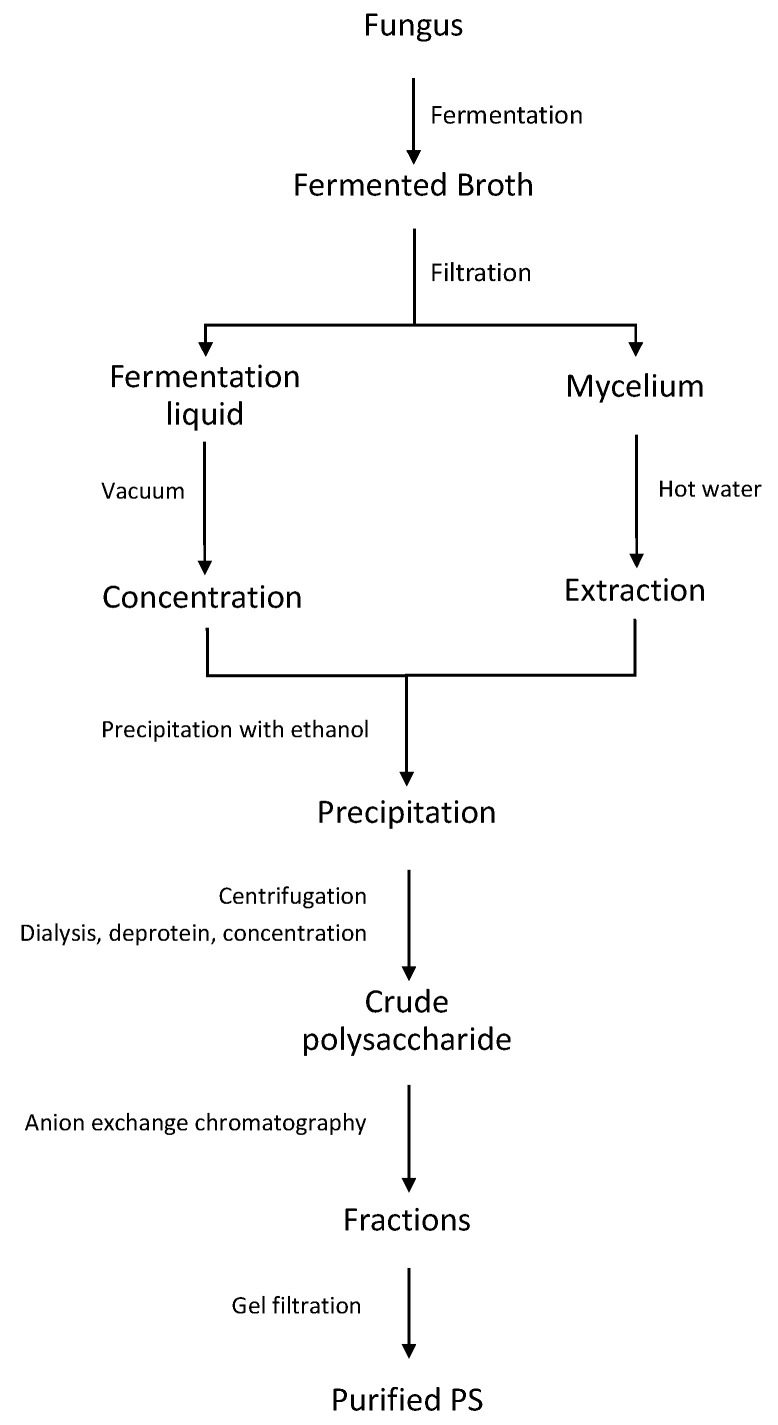
Scheme for the isolation and purification of exopolysaccharides produced by marine fungi.

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
