# Peer review of "Antioxidant Molecules from Marine Fungi: Methodologies and Perspectives"

_antioxidants, 2020, doi:10.3390/antiox9121183_

Round 1
Reviewer 1 Report
This is an interesting review article concerning antioxidant molecules from marine fungi, which could be seen as a promising source of new compounds since the marine environment covers 70% of the planet and is still poorly explored.
The review is very comprehensive and well written.
According to my opinion, the manuscript should be accepted for publication in Antioxidants, after minor corrections.
The comments are as follows:
- The Abstract seems to be too long, with more than 300 words and is suggested to be shortened.
- Lines 124-125: ROS in many cases acts just like oxidants, meaning they oxidize other molecules, for example, biomolecules, by accepting the biomolecule`s electron, with no transfer of oxygen atom. I propose to change the sentence as follow: that have the tendency to accept an electron to other substances
- Line 139: It should be considered that radical termination is determined by the rate constant of the termination reaction and the concentration of radicals, which are kinetic rather than solely thermodynamic factors.
- In the paragraph starting with line 146, the lines of defense of antioxidants are not clearly explained, the difference among levels.
- Line 166 – vitamin E reactions besides proton include electron transfer.
- Line 178 – HAT reaction mechanism includes electron and proton transfer, and in that case, the product of this step is not antioxidant radical, but neutral molecule (since antioxidant loses one electron and one proton i.e. one negative and one positive charge)
- Line 205, should be considered references Foti et al Organic Letters 2011, and Ingold Chemical Reviews 2014 which doubt the DPPH test.
Author Response
Naples, Nov 20th, 2020
Dear Editor,
Thank you very much for your consideration of the article entitled: “Antioxidant molecules from marine fungi: methodologies and perspectives” written by Giovanni Andrea Vitale, Daniela Coppola, Fortunato Palma Esposito, Carmine Buonocore, Janardhan Ausuri, Emiliana Tortorella and myself
The revised version contains all the constructive, up to point modifications requested by the Editor and Reviewers. In addition to the revised manuscript a point by point response is included.
REVIEWER 1
- The Abstract seems to be too long, with more than 300 words and is suggested to be shortened.
Answer: Thanks for your comment, we shortened the abstract how required, we hope now it is more concise.
- Lines 124-125: ROS in many cases acts just like oxidants, meaning they oxidize other molecules, for example, biomolecules, by accepting the biomolecule`s electron, with no transfer of oxygen atom. I propose to change the sentence as follow: that have the tendency to accept an electron to other substances.
Answer: We changed the sentence as suggested by you.
- Line 139: It should be considered that radical termination is determined by the rate constant of the termination reaction and the concentration of radicals, which are kinetic rather than solely thermodynamic factors.
Answer: Thank you for the comment, we fixed this point.
- In the paragraph starting with line 146, the lines of defense of antioxidants are not clearly explained, the difference among levels.
Answer: Thanks for your comment, effectively it was too strict, we gave an extensive explanation for each level.
- Line 166 – vitamin E reactions besides proton include electron transfer.
Answer: In this case we meant it as an example of an antioxidant mechanisms involving vitamin E not as the only one, on this purpose we wrote “for instance”.
- Line 178 – HAT reaction mechanism includes electron and proton transfer, and in that case, the product of this step is not antioxidant radical, but neutral molecule (since antioxidant loses one electron and one proton i.e. one negative and one positive charge)
Answer: The molecule which the hydrogen and the electron are extracted from becomes a free radical itself, as it is described in the general HAT reaction where A1-H becomes A1 · : A1–H + A2 · → A1 · + A2–H
(Mayer, J. Am. Chem. Soc. 2007, 5153)
- Line 205, should be considered references Foti et al Organic Letters 2011, and Ingold Chemical Reviews 2014 which doubt the DPPH test.
Answer: Thanks for your comment, we added the above cited references.
REVIEWER 2
- This review of antioxidants from marine funghi sources will be an important work after it has been carefully revised and finalized. Currently, it is still quite sketchy and unfinished. Especially, the English language needs to be carefully checked but I also recommend the authors to consult a chemist to verify that all the chemistry described in the paper is correct. Also, the text requires a number of figures to illustrate the contents in many places. For example, it would be useful for the reader if the chemistry behind the in vitro assays was illustrated with chemical formula and reactions. Also, examples of the many different kinds of compounds described in the paper should be given by showing their chemical formula.
Answer: Thank you for your comments, they have been useful to improve this manuscript, we carefully revised the paper with particular attention to the English.
With regard to the antioxidant assays, there are several manuscript we cited where the chemistry behind these assays is depicted, that is not our aim, which was to synthetize the commonly used in vitro assays in the research of new antioxidants.
Concerning your suggestion to find a way to better illustrate the different classes of metabolites herein discussed, we created a new figure (Figure 1, subchapter 3.1) where we drew for each chemical class the structure of a representative metabolite.
REVIEWER 3
- Abbreviations in the abstract (i.e., “ROVs”, “ROS”, and “RNS”) are not needed because they appear only once.
Answer: We removed those abbreviations.
- Abbreviations in tables, such as DPPH and FRAP in Table 1, are usually explained in their footnotes if they are used.
Answer: Thank you for this specification, we put footnotes for those abbreviations.
- “EPS” in the title of Figure 1 should be spelled out.
Answer: We fixed this point
Now, we do hope that the manuscript is acceptable for publication.
Thank you in advance for your kind cooperation,
Yours sincerely
Reviewer 2 Report
This review of antioxidants from marine funghi sources will be an important work after it has been carefully revised and finalized. Currently, it is still quite sketchy and unfinished. Especially, the English language needs to be carefully checked but I also recommend the authors to consult a chemist to verify that all the chemistry described in the paper is correct. Also, the text requires a number of figures to illustrate the contents in many places. For example, it would be useful for the reader if the chemistry behind the in vitro assays was illustrated with chemical formula and reactions. Also, examples of the many different kinds of compounds described in the paper should be given by showing their chemical formula.
Author Response

(The authors gave the same response as above.)

Reviewer 3 Report
The authors reviewed articles regarding antioxidants produced by marine fungi. Wide-ranging issues, including the deactivation mechanisms of reactive species by antioxidants, assays to detect them, and molecules with antioxidant activity extracted from marine fungi, are comprehensively covered.
This is an interesting article taking up topics of compounds that may be extracted from abundantly present marine fungi. It will contribute to the progress in exploiting useful natural compounds. The manuscript is well written, and I enjoyed reading it.
Although I do not have any critical comments, minor issues to strengthen this manuscript are raised as follows:
- Abbreviations in the abstract (i.e., “ROVs”, “ROS”, and “RNS”) are not needed because they appear only once.
- Abbreviations in tables, such as DPPH and FRAP in Table 1, are usually explained in their footnotes if they are used.
- “EPS” in the title of Figure 1 should be spelled out.
Author Response

(The authors gave the same response as above.)
